# Treatment of Dystrophic *mdx* Mice with an ADAMTS-5 Specific Monoclonal Antibody Increases the Ex Vivo Strength of Isolated Fast Twitch Hindlimb Muscles

**DOI:** 10.3390/biom10030416

**Published:** 2020-03-07

**Authors:** Alex B. Addinsall, Leonard G. Forgan, Natasha L. McRae, Rhys W. Kelly, Penny L. McDonald, Bryony McNeil, Daniel R. McCulloch, Nicole Stupka

**Affiliations:** 1Centre for Molecular and Medical Research, School of Medicine, Deakin University, Waurn Ponds, VIC 3216, Australia; alex.addinsall@ki.se (A.B.A.); leonard.forgan@deakin.edu.au (L.G.F.); natasha.mcrae1@deakin.edu.au (N.L.M.); rwkelly@deakin.edu.au (R.W.K.); penny.mcdonald@mcri.edu.au (P.L.M.); bryony.mcneill@deakin.edu.au (B.M.); daniel.mcculloch@uq.net.au (D.R.M.); 2Department of Physiology and Pharmacology, Karolinska Insitutet, 171 77 Stockholm, Sweden; 3Department of Medicine—Western Health, The University of Melbourne, St. Albans, VIC 3021, Australia; 4Australian Institute for Musculoskeletal Science (AIMSS), 176 Furlong Road, St. Albans, VIC 3021, Australia

**Keywords:** ADAMTS-5, contractile function, desmin, Duchenne muscular dystrophy, *mdx* mouse, myogenesis, skeletal muscle, versican, versikine

## Abstract

Aberrant extracellular matrix synthesis and remodeling contributes to muscle degeneration and weakness in Duchenne muscular dystrophy (DMD). ADAMTS-5, a secreted metalloproteinase with catalytic activity against versican, is implicated in myogenesis and inflammation. Here, using the *mdx* mouse model of DMD, we report increased ADAMTS-5 expression in dystrophic hindlimb muscles, localized to regions of regeneration and inflammation. To investigate the pathophysiological significance of this, 4-week-old *mdx* mice were treated with an ADAMTS-5 monoclonal antibody (mAb) or IgG2c (IgG) isotype control for 3 weeks. ADAMTS-5 mAb treatment did not reduce versican processing, as protein levels of the cleaved versikine fragment did not differ between hindlimb muscles from ADAMTS-5 mAb or IgG treated *mdx* mice. Nonetheless, ADAMTS-5 blockade improved ex vivo strength of isolated fast *extensor digitorum longus*, but not slow *soleus*, muscles. The underpinning mechanism may include modulation of regenerative myogenesis, as ADAMTS-5 blockade reduced the number of recently repaired desmin positive myofibers without affecting the number of desmin positive muscle progenitor cells. Treatment with the ADAMTS-5 mAb did not significantly affect markers of muscle damage, inflammation, nor fiber size. Altogether, the positive effects of ADAMTS-5 blockade in dystrophic muscles are fiber-type-specific and independent of versican processing.

## 1. Introduction

Duchenne Muscular Dystrophy (DMD) is an X-linked, pediatric disease with devastating effects on skeletal, respiratory and cardiac muscles. Patients are commonly wheelchair-bound by 12 years of age and succumb to cardiorespiratory failure by the third decade of life. DMD is caused by mutations in the dystrophin gene [1,2,3]. Dystrophin links the contractile apparatus to the extracellular matrix (ECM) and provides structural support to the sarcolemma during muscle contraction. The loss of dystrophin and the dystrophin associated protein complex (DAPC) renders dystrophic muscle highly prone to contraction-induced damage [4]. Chronic muscle degeneration combined with a heightened pro-inflammatory state, compromise muscle repair, leading to muscle loss and expansion of the ECM (fibrosis) [5,6].

Fibrosis is typically considered to be a hallmark of a developed pathology. However, in DMD endomysial matrix expansion precedes overt muscle degeneration and is observed in patients as young as 2.5 weeks of age [7]. This expansion of the endomysial matrix is thought to actively contribute to the degeneration of dystrophic muscles by heightening inflammation and compromising regenerative myogenesis [8,9,10]. Studies in vertebrate models with a high capacity for tissue repair without fibrosis, such as Urodele amphibians, have shown that effective regenerative myogenesis depends on carefully regulated ECM synthesis and remodeling [11]. Following injury, there is a rapid shift from a stiff collagen- and laminin-rich mature matrix to a softer transitional matrix enriched in versican and hyaluronan. This transitional matrix modulates the behavior of tissue progenitor cells, inflammatory cells and fibroblasts through mechanical and biochemical signals, which include the regulation of growth factor and cytokine bioavailability [12]. Successful regeneration also encompasses transitional matrix remodeling by various ECM proteases, including ADAMTS metalloproteinases with catalytic activity against versican, followed by the re-deposition of a mature matrix [11,13]. The proteolytic processing of transitional matrix proteins generates bioactive peptide fragments, which may also regulate cellular processes relevant to muscle regeneration and degeneration in dystrophy. For example, V0/V1 versican processing by ADAMTS versicanases generates the bioactive versikine fragment, which, depending on its biological context may stimulate apoptosis [13], inflammation [14] or proliferation [15].

Fibrosis in dystrophic muscles from patients with DMD and *mdx* mice (the murine model of DMD) is characterized by the upregulation of mature and provisional matrix proteins and proteases, including ADAMTS-5, V0/V1 versican, and the catalytically processed versikine fragment [10,16,17,18,19,20]. This chronic pro-fibrotic state leads to aberrant growth factor and cytokine signaling (including TGFβ), excess inflammation, failed myogenesis, and further matrix expansion. To date, the pathophysiological implications of dysregulated provisional matrix synthesis and remodeling in DMD are not well recognized. Despite extensive pre-clinical research, there is no effective therapeutic strategy to ameliorate fibrosis in dystrophy. Thus, we would argue that the provisional matrix is a viable upstream target to improve the efficacy of muscle regeneration in dystrophy and to ameliorate fibrosis, with the ADAMTS and V0/V1 versican enzyme—substrate axis being of pathophysiological significance.

There is increasing recognition for a role of V0/V1 versican and ADAMTS versicanases in myogenesis. *V0/V1 versican* and *Adamts-1*, *-4*, *-5* and *-15* gene expression is increased in developing mouse hindlimb skeletal muscles and during myogenic differentiation in vitro [21]. Indeed, *Adamts-5* is highly expressed during murine limb bud myogenesis and shows overlapping expression with one of its key substrates, versican [22]. The human *ADAMTS-5* gene contains binding elements for muscle regulatory factors, which are essential for myogenic differentiation [23]. ADAMTS-15 is also highly expressed in developing limb muscles where it is co-localized to the transitional matrix, as indicated by hyaluronan staining [24]. Versican is part of the satellite cell niche [25], can stimulate myoblast proliferation [26], and during myogenic differentiation, remodeling of a versican rich pericellular matrix by ADAMTS-5 facilitates the fusion of C2C12 myoblasts into multinucleated myotubes [21]. Interestingly, ADAMTS-15 can rescue the reduction in myoblast fusion following *Adamts-5* gene knockdown, indicating redundancy in versican processing by ADAMTS versicanases during myogenesis [21]. ADAMTS-5 may also modulate myogenesis via cellular mechanisms independent of versican processing. In zebrafish embryos, *adamts-5* knockdown with morpholinos impaired somite patterning and early myogenesis due to disrupted Sonic hedgehog (Shh) signaling. This impairment was rescued with a catalytically inactive *Adamts-5* construct, suggesting a putative role for the ancillary domain of ADAMTS-5 in myogenesis [27]. 

Also relevant to the pathology of muscular dystrophy, is that versican and ADAMTS versicanases have been implicated in regulating inflammation in various disease models [28,29,30]. A carefully regulated inflammatory response is necessary for effective regenerative myogenesis. Interestingly, ADAMTS-1 released by macrophages following injury stimulates satellite cell activation [31], perhaps through versican remodeling in the satellite cell niche [25]. Versican remodeling by ADAMTS versicanases has been reported in dystrophic muscles from *mdx* mice and patients with DMD, as indicated by the co-localization of versikine to regions of regeneration and inflammation [10,17]. *Adamts-5* mRNA transcripts are upregulated in hindlimb muscles from *mdx* mice [19]. ADAMTS-5 protein is specifically upregulated in serum from patients with DMD and in *mdx* mice, where experimental evidence suggests that it may have potential as a therapy responsive biomarker [20]. It is not known whether the upregulation of ADAMTS-5 in dystrophic muscles is an adaptive or a pathological response. Increased ADAMTS-5 may facilitate regenerative myogenesis via versican processing [21] and/or catalytically independent signaling pathways (e.g., Shh) [27]. Conversely, ADAMTS-5 is closely associated with a heightened inflammatory state (best described in the context of osteoarthritis [32]). High levels of the cleaved versikine fragment may exacerbate dystrophic muscle pathology by stimulating apoptosis [13] and inflammation [14].

ADAMTS-5 is an important pharmacological target for osteoarthritis, where excess degradation of its substrate aggrecan leads to cartilage destruction. GlaxoSmithKline (GSK) developed a humanized, selective monoclonal antibody (mAb) against ADAMTS-5. This ADAMTS-5 mAb reduces aggrecan degradation via an allosteric lock mechanism. By binding to the catalytic and disintegrin-like domains of ADAMTS-5, the enzyme’s ability to engage and cleave substrate is inhibited [33], as the catalytic and disintegrin-like domains are necessary for full proteolytic activity [34]. Intraperitoneal injection (IP) of the ADAMTS-5 mAb leads to distribution within the musculoskeletal system. Its effects are relatively long lasting, as indicated by in vivo studies using mouse models of cartilage destruction where weekly treatment demonstrated biological efficacy [33]. 

Given the high level of ADAMTS-5, V0/V1 versican and versikine, it was hypothesized ADAMTS-5 blockade with the GSK ADAMTS-5 mAb would modulate the function and structure of dystrophic hindlimb muscles from young *mdx* mice. Thus, juvenile male and female *mdx* mice were treated from 4 to 7 weeks of age with a weekly injection of the ADAMTS-5 mAb [33]. At approximately 21 days of age, hindlimb muscles from *mdx* mice undergo a bout of degeneration, which is followed by a period of active regenerative myogenesis concurrent with postnatal skeletal muscle growth [35]. Our experimental design allowed for the examination the functional and structural effects of the ADAMTS-5 blockade on muscle regeneration. The effects on early muscle degeneration could not be assessed, as this occurred one week prior to the onset of treatment.

## 2. Materials and Methods 

### 2.1. Ethics Approval, Mouse Husbandry and Antibody Treatment

Mouse studies were approved by the Animal Ethics Committees at Deakin University (G35-2013) and at La Trobe University (AEC16-08). Animal care and experimental procedures were conducted in accordance with the Australian Code of Practice for the Care and Use of Animals for Scientific Purposes. All mice were maintained on an alternating 12 h light and 12 h dark cycle, with standard mouse chow and water provided ad libitum.

Male *mdx* mice and C57BL/10 mice, 3 to 6 months of age, were used for immunohistochemical characterization of ADAMTS versicanases (ADAMTS-1, -5 and -15) and versican remodeling in dystrophic hindlimb muscles, unless otherwise indicated in the figure legends. Mice were deeply anaesthetized with sodium pentobarbitone (60 mg/kg) via an intraperitoneal injection and killed by cardiac excision. *Tibialis anterior* (TA) muscles were excised, embedded in optimal cutting temperature compound (OCT) and frozen in thawing 2-methylbutane cooled in liquid nitrogen.

To investigate the effects of ADAMTS-5 blockade on dystrophic muscle pathology, pregnant female *mdx* mice were obtained from the Animal Resource Centre (Canning Vale, Western Australia). At 4 weeks of age male and female *mdx* pups were randomized to receive an ADAMTS-5 monoclonal antibody (mAb, 12F4.1H7) or an isotype IgG2c control antibody (IgG) kindly provided by GSK. The ADAMTS-5 mAb has a KD = 0.035 nM and IC_50_ = 1.46 nM; antibody development and characterization was completed by GSK, as described previously [33]. In accordance with the published literature and GSK recommendations, the ADAMTS-5 mAb or the IgG control antibody were administered once weekly for three weeks at a dose of 10 mg/kg body weight via intraperitoneal injections [36]. Following the onset of treatment, mice were monitored and weighed every second day until the conclusion of the study at 7 weeks of age. 

### 2.2. Ex Vivo EDL and Soleus Contractile Function Testing

At 7 weeks of age, *mdx* mice were anaesthetized via an intraperitoneal injection of medetomidine (0.6 mg/kg), midazolam (5 mg/kg) and fentanyl (0.05 mg/kg), such that they were unresponsive to tactile stimuli. Isometric contractile properties of isolated fast twitch EDL and slow twitch *soleus* hind limb muscles were evaluated ex vivo, as described in detail previously [37,38]. Briefly, EDL and *soleus* muscles were tied at the proximal and distal tendons with braided surgical silk, surgically excised, and transferred to the 1300A Whole Mouse Test System (Aurora Scientific) organ bath filled with Krebs Ringer solution (137 mM NaCl, 24 mM NaHCO_3_, 11 mM D-glucose, 5 mM KCl, 2 mM CaCl_2_, 1 mM NaH_2_PO_4_H_2_O, 1 mM MgSO_4_, 0.025 mM d-tubocurarine chloride), bubbled with Carbogen (5% CO_2_ in O_2_; BOC Gases), and maintained at 25 °C. The distal tendon of the muscle was tied to an immobile pin and the proximal tendon was attached to the lever arm of a dual mode force transducer (300-CLR; Aurora Scientific). EDL and *soleus* muscles were stimulated by supramaximal square wave pulses of 350 ms and 1200 ms in duration, respectively, delivered by two platinum electrodes that flanked the length of the muscle. All stimulation parameters and contractile responses were controlled and measured using Dynamic Muscle Control Software (DMC v5.415), with on board controller interfaced with the transducer control/feedback hardware (Aurora Scientific). 

A series of 1 Hz isometric twitch contractions were used to determine optimal muscle length (L_o_). Following 4 min of rest, the maximum isometric tetanic force (P_o_) production was determined from the plateau of the force frequency curve (FFC). The EDL muscles were stimulated at 10, 30, 50, 60, 80, 100 and 120 Hz and *soleus* at 10, 20, 30, 50, 60, 80 and 100 Hz, with 2 min rest between stimulations. Muscles were again rested for 4 min. To assess tolerance to repeated contractile activity and as an indicator of fatigability, muscles were stimulated sub-maximally (60 Hz) once every 5 s for 4 min [38]. Recovery of P_o_ was determined by stimulating the muscles at 60 Hz at 2, 5, and 10 min post fatigue testing. 

Following the completion of contractile function testing, muscles were trimmed of tendons, weighed and snap frozen in liquid nitrogen for gene expression analysis. Contralateral hindlimb muscles were also excised, embedded in OCT, and frozen in thawing 2-methylbutane. Muscle cross-sectional area was determined by dividing the muscle mass by the product of optimum fiber length (L_f_) and 1.06 mg·mm^−3^, the density of mammalian muscle. L_f_ was determined by multiplying L_o_ by previously determined L_o_/L_f_ ratios; 0.44 for the EDL and 0.71 for the *soleus* [39]. Since P_t_ and P_o_ are dependent upon muscle size, these values were normalized for muscle cross-sectional area and expressed as specific force (sP_t_ and sP_o_; mN·mm^−2^) for the force frequency curve. To assess fatiguability and force recovery, data from the fatigue protocol were normalized to the first 60 Hz contraction and expressed as a percentage.

### 2.3. Immunohistochemistry for ADAMTS-1, -5 and -15 and Versikine

Transverse frozen sections were cut from the mid-belly of TA or EDL muscles at a thickness of 8 μm. To assess the co-localization of ADAMTS-5 with CD68 positive macrophages or desmin positive myoblasts and newly regenerated myotubes, serial sections were cut from *mdx* TA muscles. Serial sections were used for the co-localization experiments because the anti-ADAMTS-5, anti-CD68 and anti-desmin antibodies were all raised in the same species (rabbit). To confirm co-localization based on tissue morphology phase images were captured and overlaid with the corresponding fluorescent images.

Immunohistochemistry for ADAMTS-1 (Origene; TA317919), ADAMTS-5 (Affinity Bioreagents; PAI-1751A), ADAMTS-15 (Abcam; ab45047), versikine (anti-DPEAAE neo-epitope; Thermo Fisher Scientific, PA1-1748A), or desmin (Abcam, ab15200) was performed as previously described [13,21]. Nuclei were counter-stained with DAPI. For EDL muscle cross-sections reacted with the anti-desmin antibody, wheat germ agglutinin (WGA) Alexa Fluor 555 (Thermo Fisher Scientific, W32464) was used as an additional stain for regions of ECM and fibrosis [40].

To assess ADAMTS versicanase immunoreactivity and for the ADAMTS co-localization experiments, for each TA muscle cross-section four representative digital images were captured with a confocal microscope at 600× magnification (Olympus; Fluoview FV10i). To quantify the number of desmin positive myoblasts and newly regenerated myofibers [41,42], due to the small cross-sectional area of EDL muscles from 7-week-old *mdx* mice, only two representative digital images were captured with a confocal microscope (Olympus; Fluoview FV10i) at 200× magnification.

### 2.4. Immunoblotting for Versikine—A Read out of Total ADAMTS Versicanase Activity

EDL and *soleus* muscles were homogenized in radio immunoprecipitation assay buffer (RIPA; Merck Millipore) with protease and phosphatase inhibitors (Thermo Fisher Scientific). Total protein content was determined using a BCA protein assay (Thermo Fisher Scientific), and 7.5 µg of unfractionated muscle homogenate was separated on a 4%–15% gradient TGX Stain-Free™ criterion gel (Bio-Rad) at 100 V. Following which the gel was activated and proteins visualized using the Chemidoc™ XRS system (Bio-Rad). Proteins were then transferred to PVDF membranes using a Turbo Blot Transfer System (Bio-Rad) at 2.5 A and 25 V for 12 min. Immunoblotting for versikine (anti-DPEAAE neo-epitope; Thermo Fisher Scientific, PA1-1748A) was performed as previously described [17]. Blots were imaged using ECL chemiluminescence. Band densitometry was performed on the western blots and the stain free gels, to confirm even loading, using Image Lab software (Bio-Rad). Versikine protein expression was normalized to the optical density of the total protein per lane on the TGX-stain free protein gel.

### 2.5. Histological Assessment of mdx EDL Muscle Morphology Following Adamts-5 mAb Treatment

As improvements in contractile function were only observed in EDL muscles, these were selected for further histological and biochemical analyses of dystrophic pathology. Transverse 8 μm thick frozen sections, cut from the mid-belly of EDL muscles from *mdx* mice treated with the ADAMTS-5 mAb or the IgG control, were stained with hematoxylin and eosin (H&E) staining to assess muscle morphology [38]. Digital images of H&E stained muscle were captured at 200x magnification (Leica; DM1000 upright microscope). All image analysis was completed using Image-Pro Plus software (Media Cybernetics). To assess muscle fiber size (minimal Feret’s diameter) and the percentage of centrally nucleated fibers, 116 ± 7 muscle fibers were quantified per cross-section per mouse. Muscle degeneration was assessed by manually circling regions of fibrosis and mononuclear infiltrate, which is composed of muscle progenitor cells, inflammatory cells and fibroblasts [43] and expressing these regions as a percentage of the total muscle cross section [37]. 

### 2.6. Real Time Quantitative PCR (qPCR)

To assess the effects of the ADAMTS-5 blockade on *V0* and *V1 versican* (*Vcan*) mRNA transcripts and gene markers of inflammation and myogenesis, whole EDL muscles were homogenized in Tri-Reagent^®^ solution (Ambion Inc, Thermo Fisher Scientific). Total cellular RNA was extracted and purified using a RNeasy^®^ Mini Kit (Qiagen). An iScript cDNA synthesis kit (Bio-Rad) was used to reverse transcribe 0.25 μg of total RNA. Quantitative RT-PCR was performed using IQ SYBR Green Super mix (Bio-Rad) and oligonucleotide primers for the genes of interest (Table 1). cDNA concentrations were determined using Quant-iT™ OliGreen^®^ ssDNA reagent (Thermo Fisher Scientific), and Ct values were normalized to cDNA content.

### 2.7. Statistical Analyses

All data are presented as MEAN ± S.E.M. Data were assumed to be normally distributed. An independent sample t-test or 2-way General Linear Model (GLM) ANOVA, followed by Tukey’s post hoc analysis where appropriate, were performed as indicated. All statistical analyses were performed using Minitab statistical software (v17), with *P* < 0.05 being statistically significant. 

## 3. Results

### 3.1. Increased ADAMTS-5 Immunoreactivity in Dystrophic mdx Compared to Wild Type Hindlimb Muscles

ADAMTS-1, -5 and -15 process versican to generate the bioactive versikine fragment [44]. ADAMTS-1, -5 and -15 immunoreactivity was very low in TA muscles from adult C57BL/10 mice (Figure 1D–F, Figure 1J–L and Figure 1P–R). These ADAMTS versicanases have a putative role in myogenesis during muscle development and regeneration [21]. In healthy, adult TA muscles where there is essentially no regeneration, it was unsurprising that ADAMTS-1, -5 and -15 protein expression was low. Whereas, in dystrophic TA muscles from *mdx* mice where there is ongoing muscle degeneration and repair, ADAMTS-5 was the most highly upregulated of the ADAMTS versicanases assessed (Figure 1G–I, Figure 1S; *P* < 0.05); although, some ADAMTS-1 and -15 immunoreactivity was also observed in dystrophic TA muscles (Figure 1A–C and Figure 1M–O). ADAMTS-5 was localized to the endomysium (white arrow) and to regions of mononuclear infiltrate (white arrowhead; Figure 1G–I). Fibroblasts [45], inflammatory cells [28,30], muscle progenitor cells and differentiating myoblasts [21,22] can all express *Adamts-5*. Using serial TA muscle cross-section from *mdx* mice, the upregulation of ADAMTS-5 in regions of regeneration and inflammation was confirmed, as indicated by the co-localization with desmin positive muscle cells (Figure 2A–D) and a close association with CD68 positive macrophages (Figure 2E–H). The relevance of the ADAMTS-5—versican enzyme—substrate axis to dystrophic muscle pathology is supported by the observations that versikine is also co-localized with newly regenerated desmin positive muscle fibers (Appendix A) and infiltrating macrophages (Appendix A) in TA muscles from *mdx* mice. 

### 3.2. Effects of ADAMTS-5 Blockade on Post-natal Growth and Muscle Mass

ADAMTS-5 blockade with the mAb was well tolerated and did not appear to compromise postnatal growth. The increase in body weight (g) during the three weeks of treatment did not significantly differ between *mdx* mice treated with the ADAMTS-5 mAb or the IgG control (Figure 3). EDL, but not *soleus*, muscles from *mdx* mice treated with the ADAMTS-5 mAb were heavier than EDL muscles from *mdx* mice treated with the IgG control. However, when EDL or *soleus* muscle mass was normalized to body weight, ADAMTS-5 blockade had no significant effect on proportional muscle size (Table 2 and Table 3, respectively). 

### 3.3. ADAMTS-5 Blockade Does Not Reduce Versican Processing in Dystrophic EDL and Soleus Muscles

It was hypothesized that ADAMTS-5 blockade would reduce versican processing and thus decrease versikine protein content [46]. Unexpectedly, versikine was readily detected in EDL muscle cross-sections from both ADAMTS-5 mAb and IgG treated *mdx* mice. In concordance with previously published findings, immunoreactivity was localized to the endomysium and to myonuclei (Figure 4A,B) [17]. When quantitatively assessed using immunoblotting, versikine protein content in either EDL or *soleus* muscles did not significantly differ between *mdx* mice treated with the ADAMTS-5 mAb or the IgG control (Figure 4C,D). This lack of difference in versikine protein levels following 3 weeks of ADAMTS-5 mAb blockade was unexpected, hence mRNA transcript abundance of the *V0* and *V1 Versican* (*Vcan*) was assessed using qRT-PCR. Overall, mRNA transcript abundance of *V0* and *V1 Vcan* was increased in EDL muscles from *mdx* mice treated with the ADAMTS-5 mAb (Figure 4E; P = 0.01). This suggests that in hindlimb muscles from young *mdx* mice, ADAMTS-5 blockade may result in a compensatory upregulation of substrate, such that versikine levels are not decreased with treatment. Thus, any observed effects of the ADAMTS-5 blockade on dystrophic muscle function and structure are likely to be independent of versican processing by ADAMTS-5.

### 3.4. Contractile Properties of Isolated, Dystrophic EDL and Soleus Muscles Following ADAMTS-5 Blockade

The ex vivo 1 Hz twitch force (P_t_), time to peak tension (TPT), and ½ relaxation time (½ RT) did not significantly differ between EDL or *soleus* muscles from *mdx* mice treated with ADAMTS-5 mAb or the IgG control. When twitch force (P_t_) was normalized to muscle cross-sectional area, ADAMTS-5 blockade increased the force output (sP_t_) in response to 1 Hz stimulation in EDL (*P* < 0.05), but not *soleus* muscles (Table 2 and Table 3, respectively). This indicates that ADAMTS-5 blockade may specifically increase muscle strength in fast dystrophic hindlimb muscles. Whilst having no significant effect on muscle relaxation or Ca^2+^ regulation in either fast or slow dystrophic hindlimb muscles, as TPT and ½ RT are determined by myosin heavy chain composition, Ca^2+^ sensitivity, and SERCA isoform and function [47].

In concordance with the 1 Hz twitch data in EDL muscles, ADAMTS-5 blockade increased the isometric strength normalized to muscle cross-sectional area (sP_o_), as indicated by an upward shift in the 1 to 120 Hz force frequency curve (*P* < 0.001; Figure 5A). Whereas, in slow *soleus* muscles, ADAMTS-5 blockade had no significant effect on isometric muscle strength (sP_o_; Figure 5C). Following assessment of isometric strength, muscle endurance was assessed. During the 4 min of intermittent, submaximal stimulation at 60 Hz, EDL muscles from *mdx* mice treated with the ADAMTS-5 mAb fatigued less than IgG control treated littermates (*P <* 0.05), and force recovery was enhanced (*P <* 0.05; Figure 5B). In *soleus* muscles, ADAMTS-5 blockade had no significant effect on the rate of fatigue or force recovery (Figure 5D). Altogether, these data demonstrate that the positive effect of ADAMTS-5 blockade on the contractile function of dystrophic hindlimb muscles is fiber type specific. 

### 3.5. Effects of ADAMTS-5 Blockade on EDL Muscle Structure

To gain insight into the potential mechanism underpinning the increase in EDL muscle strength following treatment with the ADAMTS-5 mAb, a morphometric analysis of H&E stained EDL muscle cross-sections was undertaken. In dystrophic EDL muscles, muscle fiber size, as indicated by min Feret’s diameter, did not significantly differ between mice treated with ADAMTS-5 mAb or the IgG control (Figure 6A). This is in concordance with a lack of effect of ADAMTS-5 blockade on the EDL muscle mass to bodyweight ratio (Table 2). As a marker of muscle damage and repair, centrally nucleated fibers were assessed in ADAMTS-5 mAb or the IgG control treated EDL muscles. The percentage of centrally nucleated fibers also did not differ between ADAMTS-5 mAb or IgG control treated mice (Figure 6B). Muscle degeneration, comprising of connective tissue and mononuclear infiltrate, also did not differ between ADAMTS-5 mAb or the IgG control treated mice (Figure 6C). In line with the morphometric analysis of muscle degeneration, *Mcp-1* and *Tgfβ1* gene expression was also not significantly altered by ADAMTS-5 blockade (*P* = 0.331 and *P* = 0.233, respectively, Figure 6D,E). Therefore, the increase in EDL muscle strength in *mdx* mice treated with the ADAMTS-5 mAb is unlikely to be mediated by muscle hypertrophy or decreased muscle degeneration. 

### 3.6. Effect of ADAMTS-5 Blockade on Markers Regenerative Myogenesis in EDL Muscles

Desmin positive muscle progenitor cells (outlined with a white circle) and small, centrally nucleated desmin positive muscle fibers (white arrow) were observed in EDL muscle cross-sections from *mdx* mice treated with either the ADAMTS-5 mAb or the IgG control (Figure 7A–D). However, the number of desmin positive muscle fibers per mm^2^ of tissue cross-section was lower in *mdx* mice treated with the ADAMTS-5 mAb (*P* < 0.05), whilst the number of desmin positive muscle progenitor cells did not significantly differ between treatment groups (Figure 7E). It should be noted that not all centrally nucleated fibers had high levels of desmin immunoreactivity (Figure 7C; white asterisks), highlighting the limitation in central nuclei as a marker of recent regeneration. A non-significant increase in the mRNA transcript abundance of the myogenic regulatory factors *Myf5* and *Myogenin* was also observed in EDL muscles from *mdx* mice treated with the ADAMTS-5 mAb (Figure 7D,E; P = 0.140 and *P* = 0.116, respectively). The desmin immunohistochemistry, and perhaps the myogenic regulatory factor gene expression data, offer preliminary *in vivo* evidence that ADAMTS-5 blockade may have the potential to modulate regenerative myogenesis in dystrophic muscles. 

## 4. Discussion

Here we report that ADAMTS-5 is highly expressed in dystrophic hindlimb muscles from *mdx* mice and upregulated in regions of regeneration and inflammation. The co-localization of ADAMTS-5 with desmin positive muscle cells in dystrophic hindlimb muscles aligns with previous observations by our laboratory that versikine, the bioactive product of versican remodeling by ADAMTS versicanases, is also co-localized with newly regenerated desmin positive muscle fibers and muscle progenitor cells in dystrophic muscles [17]. The immunoreactivity of ADAMTS-5 in regions of mononuclear infiltrate and inflammation are also consistent with the co-localization of versikine and infiltrating macrophages in dystrophic muscle [10]. Reports that macrophages express *Adamts-5* mRNA transcripts in vitro further support our immunohistochemical data. Altogether, our findings highlight the potential relevance of ADAMTS-5 and its substrate versican to the pathogenesis of DMD. 

When young *mdx* mice were treated with an ADAMTS-5 mAb for 3 weeks beginning at 4 weeks of age, the isometric strength and endurance of dystrophic hindlimb muscles was improved in a fiber type specific manner. The positive effects of ADAMTS-5 blockade on the contractile function of EDL muscles were independent of a reduction in versican remodeling, as versikine protein levels in EDL muscles did not differ between *mdx* mice treated with the ADAMTS-5 mAb or the IgG control. The ADAMTS-5 mAb used in this study has a high binding affinity and a slow off rate, which makes it very effective at blocking aggrecanolysis in cartilage [33]. Indeed, in human osteoarthritic cartilage explants, in cynomolgus monkey cartilage, and in a mouse model of mechanical allodynia, treatment with the GSK ADAMTS-5 mAb decreased substrate (aggrecan) degradation, demonstrating successful inhibition of ADAMTS-5 catalytic activity [33,36]. Thus, compensatory upregulation of other ADAMTS versicanases may contribute to the lack of difference in versikine levels following treatment with the ADAMTS-5 mAb. 

In this study, treatment with the ADAMTS-5 mAb coincided with a period of active regeneration in young *mdx* mice [35], and regenerative myogenesis in vivo recapitulates many of the cellular processes necessary for embryonic skeletal muscle development and myoblast differentiation in vitro. In developing embryos and cell culture models, myogenesis is associated with an upregulation of ADAMTS versicanases, with *Adamts-1, -4*, *-5* and *-15* mRNA transcripts all increased [21]. During C2C12 myoblast fusion, ADAMTS-15 can compensate for ADAMTS-5 with regards to versican processing [21]. During development, versican processing is also critical for interdigital web regression. Immediately prior to apoptosis, *Adamts-1, -5, -9* and *-20* are all upregulated in interdigital cells [13]. Compensation by other ADAMTS versicanases ensures that the incidence of soft-tissue syndactyly is low in *Adamts-5^-/-^* knockout mice; however, it reaches full penetrance in *Adamts-5, Adamts-9* and *Adamts-20* combinatorial mutants [13].

Redundancy in versican processing by other ADAMTS proteases following ADAMTS-5 blockade or deletion appears to be modulated by biological context. Indeed, at E13.5, intense versikine immunoreactivity was observed in developing hindlimb muscles from wild type and *Adamts-5^-/-^* knockout mice. Whereas, at 10 days and 3 weeks of age, versikine was detected in hindlimb muscles from wild type mice but not *Adamts-5^-/-^* knockout mice [21]. Thus, in healthy muscles compensatory versican processing by other ADAMTS versicanases may occur during developmental myogenesis, but not postnatal growth. This fits with the dramatic downregulation of *Adamts-1, -4*, *-5* and *-15* mRNA transcripts in hindlimb muscles from wild type mice during postnatal growth (encompassing 10 days to 3 weeks of age to adulthood) [21]. Whereas in dystrophic muscles, chronic muscle damage stimulates regenerative myogenesis, recapitulating cellular mechanisms of developmental myogenesis making compensatory versican processing more likely.

Increased V0/V1 versican gene transcription leading to increased substrate content could also have contributed to this lack of difference in versikine protein content following ADAMTS-5 blockade. In knee joints from healthy mice, treatment with the ADAMTS-5 mAb increased proteoglycan content, as assessed by histological staining [33]. Furthermore, Gorski et al. observed that the genetic deletion of ADAMTS-5 did not affect versican (or aggrecan) cleavage and concluded that ADAMTS-5 can have biological effects independent of substrate proteolysis [48]. Dancevic et al. identified a role for ADAMTS-5 in early myogenesis in zebrafish embryos which also was independent of its catalytic function [27].

The increase in isometric strength (sP_o_) in EDL muscles from *mdx* mice treated with the ADAMTS-5 mAb was not mediated by an increase in fiber size nor a reduction in inflammation and muscle damage, as indicated by morphometric analysis and the mRNA transcript abundance of pro-inflammatory gene markers (*Mcp-1* and *Tgfβ1*). Conversely, in a mouse model of mechanical allodynia and knee joint degeneration, treatment with the ADAMTS-5 mAb decreased Mcp-1 protein production in isolated and cultured dorsal root ganglia cells [36], indicating that biological context may determine the effects of ADAMTS-5 blockade on inflammation.

Regeneration efficacy following damage determines muscle strength (sP_o_). Treatment with the ADAMTS-5 mAb had no significant effect on the proportion of centrally nucleated fibers. However, these persist for a prolonged period following injury, and as such provide limited insight into the efficacy of regenerative myogenesis. Hindlimb muscles from young *mdx* mice undergo an acute bout of necrosis at three weeks of age, with the regenerated, centrally nucleated myofibers persisting for up to 14 weeks [49]. Studies in wild type mice using various models of muscle damage have also reported centrally nucleated fibers for up to 3 months post-injury, long after satellite cell proliferation and myoblast differentiation have ceased [50,51]. Given that *mdx* mice were treated with the ADAMTS-5 mAb between 4 and 7 weeks of age, it is impossible to discern from H&E stained EDL cross-section which centrally nucleated fibers represent repair from that initial bout of myonecrosis at 3 weeks of age and which are indicative of more recent damage and repair [35]. To begin to assess the effects of ADAMTS-5 blockade on muscle repair, desmin positive muscle progenitor cells and desmin positive, newly regenerated muscle fibers were quantified in EDL cross-sections. The reduction in the number of desmin positive muscle fibers, but not desmin positive progenitor cells, suggests that ADAMTS-5 blockade may improve the efficacy of regenerative myogenesis, which in turn would account for the increase in EDL muscle strength. This interpretation needs to be carefully interrogated in follow-up studies using more sensitive markers of regenerative efficacy. Specifically, immunohistochemical staining for muscle fibers expressing embryonic and neonatal myosin to assess initiation of regeneration at 1 to 3 days post-damage and ongoing regeneration 1 to 3 weeks post injury, respectively. This should be supported by an assessment of satellite cell proliferation and myoblast differentiation.

It remains to be determined how treatment with the ADAMTS-5 mAb improves the pathology of fast dystrophic muscles, if a reduction in versican processing is not the underpinning mechanism. Treatment with the ADAMTS-5 mAb may result in the accumulation of ADAMTS-5 with a locked catalytic and distingrin domain [33] and a free ancilliary domain which has a role in substrate recognition and tissue compartmentalization [52,53]. There is *in vivo* evidence from ‘in-frame’ *Adamts-5^-/-^* knockout mice [48] and zebrafish [27], that ADAMTS-5 with intact ancilliary and disintegrin domains has biological activity despite a loss of catalytic function. The potential role of ADAMTS-5 ancilliary and disintegrin domains in dystrophic muscle pathology warrants further investigation. Especially, if the underpinning mechanism involves regulation of Sonic hedgehog (Shh) signaling, as proposed by Dancevic et al. [27]. Shh signaling is important for effective regenerative myogenesis [54], and is compromised in fast hindlimb muscles from *mdx* mice, where it is associated with increased fibrosis [55].

## 5. Conclusions

This is the first published study to identify a therapeutic effect of ADAMTS-5 blockade in dystrophic muscles; however, there are limitations which need to be acknowledged. The treatment protocol used, whilst capturing postnatal growth, was brief and included male and female *mdx* mice. Exhibiting limited fibrosis and a high regenerative capacity, the pathology of hindlimb muscles from young *mdx* mice is mild compared to young patients with DMD [35]. It is yet to be determined whether the effects of ADAMTS-5 blockade on EDL muscle contractile function, translate to functional improvements in exercise capacity and whole-body strength (e.g., grip strength). Therefore, longer duration studies are needed in male *mdx* mice, which include an assessment of whole-body exercise performance and diaphragm muscle structure and contractile function. The effects of ADAMTS-5 blockade on diaphragm muscle contractile also warrants further investigation. In *mdx* mice, it is the diaphragm which best models DMD pathology due to high levels of endomysial fibrosis and impaired strength and endurance [56,57]. 

Altogether, our findings demonstrate an association between ADAMTS-5, versican remodeling and regenerative myogenesis in dystrophic muscles. More importantly, the observed therapeutic benefits of ADAMTS-5 blockade in EDL muscles from *mdx* mice are of pathophysiological and translational relevance, as in DMD it is the fast muscle fibers which are preferentially vulnerable to degeneration [58]. There is an urgent, unmet need for new therapeutic strategies targeted towards the extracellular matrix of dystrophic muscles, given that fibrosis is an active driver of muscle degeneration in dystrophy [5].

## Figures and Tables

**Figure 1 biomolecules-10-00416-f001:**
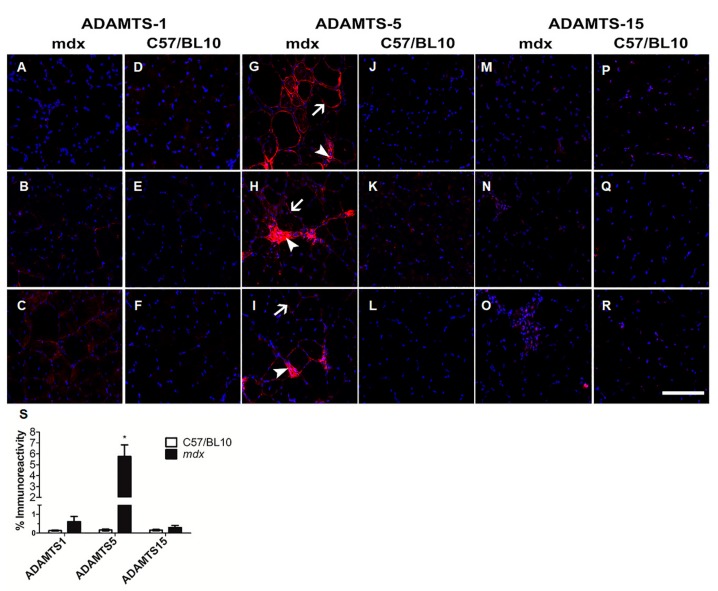
Expression of ADAMTS versicanases in healthy and dystrophic muscles. Representative images of ADAMTS-1, ADAMTS-5 and ADAMTS-15 immunoreactivity (red) and nuclei (blue). ADAMTS-1 immunoreactivity in TA muscle cross-sections from *mdx* (**A**–**C**) and C57/BL10 (**D**–**F**) mice. ADAMTS-5 immunoreactivity in TA muscle cross-sections from *mdx* (**G**–**I**) and C57/BL10 (**J**–**L**) mice, where ADAMTS-5 was found to be highly expressed in the endomysium (white arrow) and within areas of mononuclear infiltrate (white arrowhead). ADAMTS-15 immunoreactivity in TA muscle cross-sections from *mdx* (**M**–**O**) and C57/BL10 (**P**–**R**) mice. (**S**) Quantitation of ADAMTS versicanase immunoreactivity demonstrated increased *Adamts-5* expression in TA muscles from *mdx* mice. **P* < 0.05; independent t-test. N = 5 C57BL/10 mice and n = 5 *mdx* mice. Scale bar = 100 µm.

**Figure 2 biomolecules-10-00416-f002:**
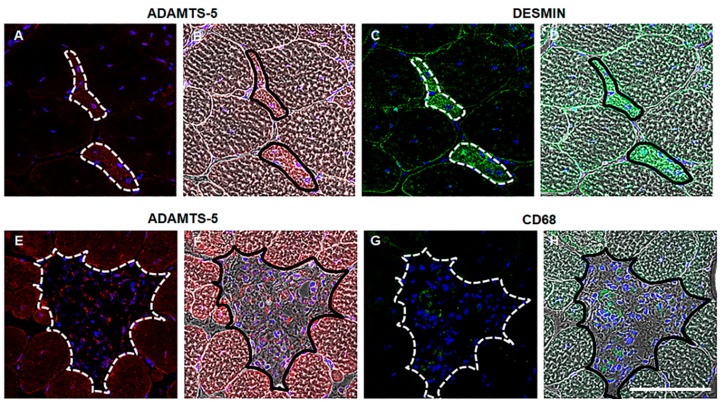
ADAMTS-5 is highly expressed in regions of regeneration and inflammation in dystrophic muscles. ADAMTS-5 immunoreactivity (red), desmin or CD68 immunoreactivity (green) and nuclei (blue) in serial cross-sections. Phase contrast images overlaid with the fluorescent signal provide morphological evidence of ADAMTS-5 and desmin or CD68 co-localization. ADAMTS-5 immunoreactivity in TA muscle cross-sections from *mdx* mice (**A** and **E**), with corresponding phase contrast image (**B** and **F**, respectively). Desmin immunoreactivity in TA muscle cross-sections from *mdx* mice (**C***),* with corresponding phase contract (**D**). CD68 immunoreactivity in TA muscle cross-sections from *mdx* mice (**G**), with corresponding phase contrast image (**H**). Regions of co-localization are indicated by the outline on the fluorescent and the phase image. N = 3 *mdx* mice. Scale bar = 100 µm.

**Figure 3 biomolecules-10-00416-f003:**
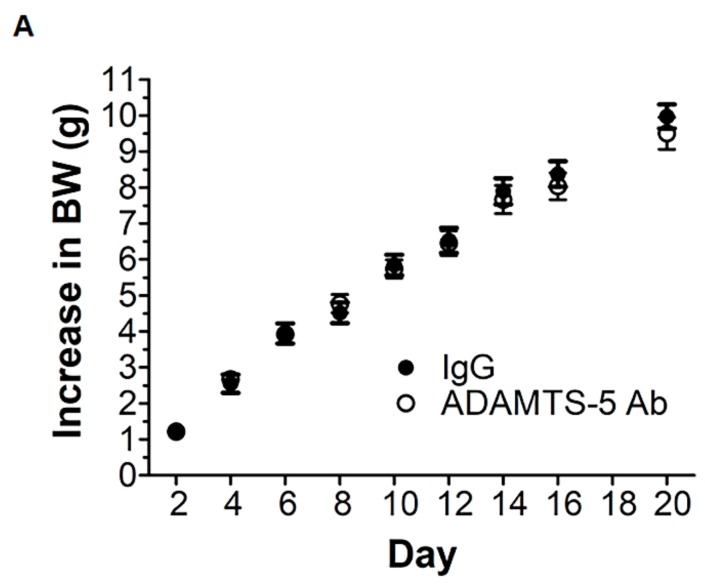
ADAMTS-5 blockade does not significantly affect postnatal growth of 4 to 7-week-old *mdx* mice. Increases in mouse body weight (g) during 20 days of treatment with the ADAMTS-5 mAb or the IgG control. N = 19–20 mice.

**Figure 4 biomolecules-10-00416-f004:**
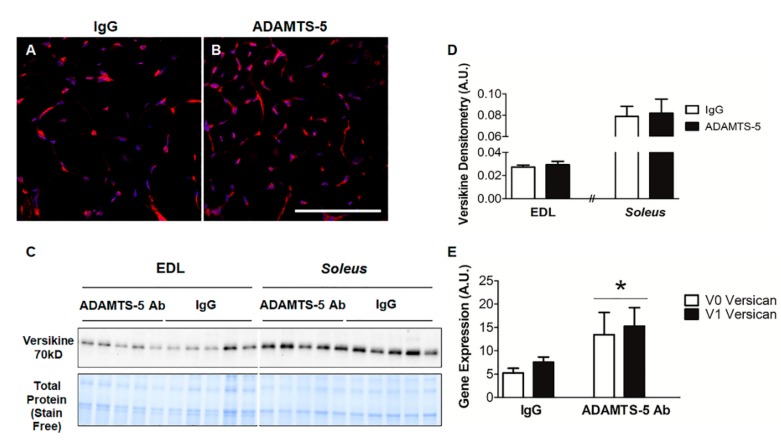
ADAMTS-5 blockade does not decrease versican processing. Representative images of versikine immunoreactivity (*red*) on EDL cross-sections from *mdx* mice treated with IgG control (**A**; n = 3) or ADAMTS-5 mAb (**B**; n = 3); nuclei (*blue*). Versikine protein levels in EDL and *soleus* muscles (**C** and **D**; n = 5 per muscle and treatment group) and (**E**) *V0* and *V1 Versican* gene expression in dystrophic EDL muscles following ADAMTS-5 blockade. **P* = 0.01, independent t-test. N = 5 mice for versikine immunoblotting and n = 13 mice for *V0* and *V1* versican gene expression. Scale bar = 100 µm.

**Figure 5 biomolecules-10-00416-f005:**
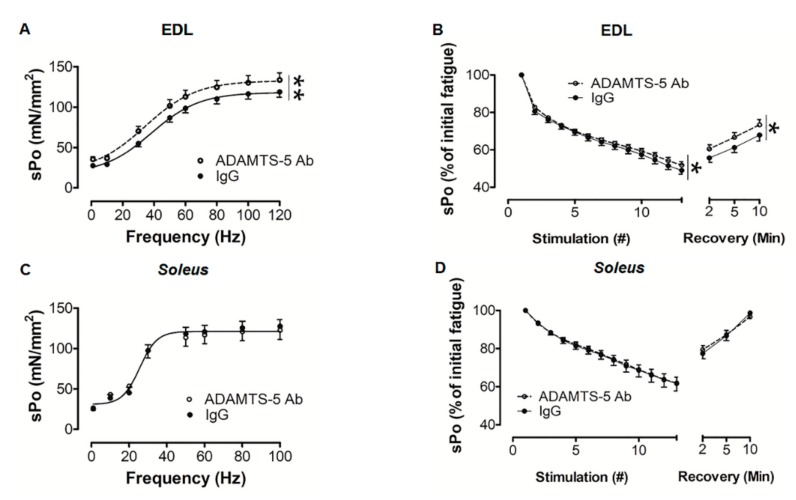
In *mdx* mice ADAMTS-5 blockade improved the contractile function of fast EDL muscles. (**A**) Treatment with the ADAMTS-5 mAb increased the specific isometric force (sP_o_) output of EDL muscles. (**B**) During 4 min of intermittent 60 Hz stimulation, the relative fatigability of EDL muscles from *mdx* mice treated with the ADAMTS-5 mAb was reduced and the relative force recovery was also improved. In *soleus* muscles, ADAMTS-5 blockade had no significant effect on (**C**) isometric strength or (**D**) fatigability and force recovery. **P* < 0.05 and ***P* < 0.001, main effect treatment, 2-way GLM-ANOVA. N = 17-19 mice.

**Figure 6 biomolecules-10-00416-f006:**
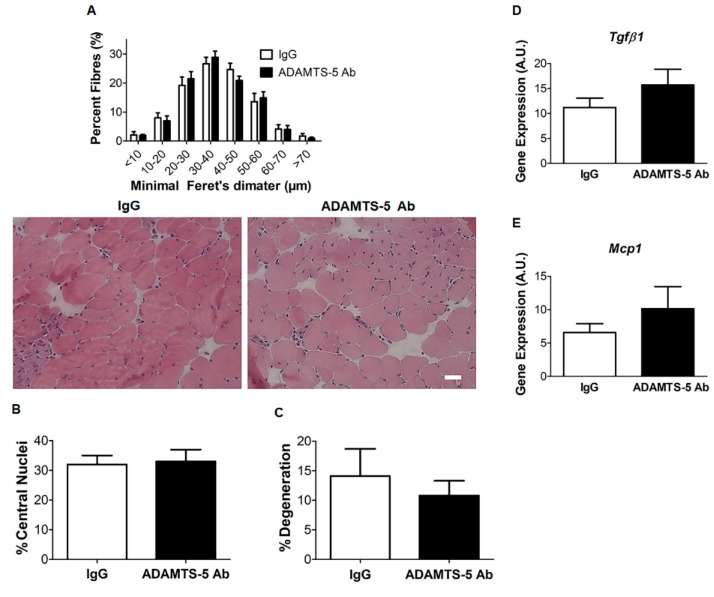
Fast twitch EDL muscle morphology is unaffected by ADAMTS-5 blockade in *mdx* mice. EDL fiber size as assessed by minimal Feret’s diameter (**A**), the proportion of centrally nucleated fibers (**B**) and percent area of degeneration and mononuclear infiltrate (**C**) did not significantly between EDL muscle cross sections from IgG and ADAMTS-5 mAb treated *mdx* mice (**A**–**C**; n = 7). The mRNA transcript abundance of *Tgfβ1* (**D**) and *Mcp-1* (**E**) did not significantly differ between EDL muscles from IgG or ADAMTS-5 mAb treated *mdx* mice. N = 7 mice for morphometric analyses and n = 13 mice for gene expression analysis. Scale bar = 100 µm.

**Figure 7 biomolecules-10-00416-f007:**
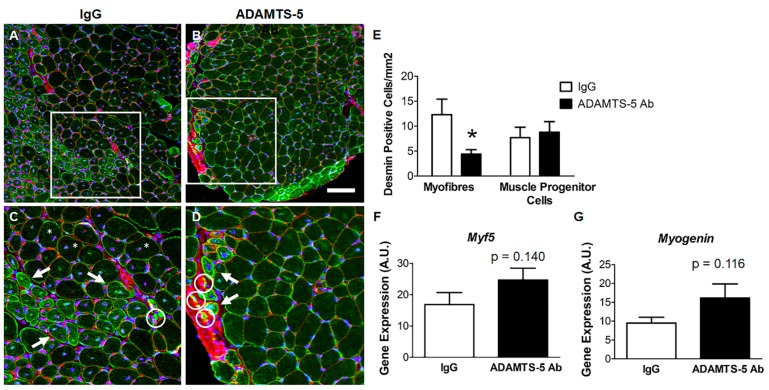
Markers of regenerative myogenesis in dystrophic EDL muscles following ADAMTS-5 blockade. Desmin immunoreactivity (green) immunoreactivity in EDL cross sections from mdx mice treated with the IgG (**A**) and the ADAMTS-5 mAb (**B**); connective tissue (red; WGA) and nuclei (blue). (**C**–**D**) Magnified area of interest (white box) with desmin positive muscle progenitor cells (white circles) and desmin positive, newly regenerated myofibers (white arrows); also, not all centrally nucleated fibers had high levels of desmin immunoreactivity (white asterisks). (**E**) Quantification of desmin positive muscle progenitor cells and regenerating myofibers. (**F**–**G**) Effects of ADAMTS-5 blockade on the mRNA transcript abundance of *Myf5* and *Myogenin*. **P* < 0.05, independent t-test. N = 8–9 mice for desmin immunoreactivity analysis and n = 13 mice for gene expression analysis. Scale bar = 100 µm.

**Table 1 biomolecules-10-00416-t001:** List of primer sequences used for quantitative RT-PCR.

Accession Number	Name	Forward Sequence	Reverse Sequence
NM_001081249.1(V0)	*V0 Vcan*	GCA GGG ACC AAG TTC CA	ATC ACT CAA TCG ACC TGT CTT GT
NM_019389.2(V1)	*V1 Vcan*	ACT GCT TTA AAC GTC GAT TGA GTG	TCA CTG CAA GGT TCC TCT
NM_011577.2	*TGFβ1*	GCC TGA GTG GCT GTC TTT TGA	CAC AAG AGC AGT GAG CGC TGA A
NM_011333.3	*MCP1*	CCCAATGAGTAGGCTGGAGA	TCTGGACCCATTCCTTCTTG
NM_031189.2	*Myogenin*	TCGGTCCCAACCCAGGA	GCAGATTGTGGGCGTCTGTA
NM_008656.5	*Myf*	CCC ACC TCC AAC TGC TCT G	CCG ATC CAC AAT GCT GGA C

**Table 2 biomolecules-10-00416-t002:** Ex-vivo EDL twitch contractile properties following ADAMTS-5 blockade.

	**Units**	**IgG**	**ADAMTS** **-5 mAb**	**P Value**
**Muscle mass**	mg	10.29 ± 0.51	*8.85 ± 0.51	0.037
**Muscle mass : BW**	mg:g	0.49 ± 0.01	0.46 ± 0.02	0.187
**P_t_**	mN	39.3 ± 3.3	48.5 ± 5.9	0.176
**sP_t_**	mN/mm^2^	27.5 ± 1.6	35.5 ± 3.0	0.026
**TPT**	s	0.2198 ± 0.0005	0.2217 ± 0.0018	0.282
**½ RT**	s	0.0151 ± 0.0006	0.0236 ± 0.0056	0.132

Data are means ± S.E.M. * Statistical significance at *P* < 0.05 for all measurements, analyzed by 2-tailed independent t-test. BW = body weight; P_t_ = twitch force; sP_t_ = specific twitch force; TPT = time to peak tension; ½ RT = half relaxation time. N = 17–19 mice.

**Table 3 biomolecules-10-00416-t003:** Ex-vivo *soleus* twitch contractile properties following ADAMTS-5 blockade.

	**Units**	**IgG**	**ADAMTS-5 mAb**	**P Value**
**Muscle mass**	mg	7.04 ± 0.46	6.71 ± 0.84	0.743
**Muscle mass : BW**	mg:g	0.34 ± 0.02	0.34 ± 0.01	0.966
**P_t_**	mN	16.1 ± 1.7	13.5 ± 1.4	0.273
**sP_t_**	mN/mm^2^	25.4 ± 2.1	25.8 ± 2.6	0.9.02
**TPT**	s	0.2661 ± 0.0303	0.2567 ± 0.0193	0.794
**½ RT**	s	0.0559 ± 0.0088	0.0646 ± 0.0219	0.725

Data are means ± S.E.M. * Statistical significance at *P* < 0.05 for all measurements, analyzed by 2-tailed independent t-test. BW = body weight; P_t_ = twitch force; sP_t_ = specific twitch force; TPT = time to peak tension; ½ RT = half relaxation time. N = 16–19 mice.

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
