# Peer review of "Treatment of Dystrophic mdx Mice with an ADAMTS-5 Specific Monoclonal Antibody Increases the Ex Vivo Strength of Isolated Fast Twitch Hindlimb Muscles"

_biomolecules, 2020, doi:10.3390/biom10030416_

Round 1
Reviewer 1 Report
Manuscript by Addinsall et al represents an important advance in the biology of the ADAMTS-5 metalloprotase and its effects on extracellular matrix synthesis in Duchenne muscular dystrophy (DMD). Authors employ an ADAMTS-5 mAb antibody to show that blockade of ADAMTS-5 influences regenerative myogenesis in dystrophic muscles. Of note, it is found that ADAMTS-5 is highly expressed in mdx mice (a murine model for DMD disease when compared with ADAMTS-1 or ADAMTS-15). Moreover, employment of those antibodies do not compromise other vital functions and versican processing to generate the bioactive fragment versikine does not seem to be affected. Greatest weakness of this work is that the molecular mechanisms underlying those effects are not explored. Taking into account the therapeutic benefits of the ADAMTS-5 blockade in the regenerative process of the muscle in DMD, it is expected that this aspect can be studied in further investigations.
Only some few minor comments:
Have the authors explored the versicanase activity of ADAMTS-1 or ADAMTS-15 in the course of their experiments? Fig 1. To better fit with the legend and the main text, pictures corresponding to ADAMTS-1 should be shown on the left part of the figure, and those corresponding to ADAMTS-15 on the right side. Fig 4C. Loading control should be indicated in both the figure and the legend. Discussion is extensive and it sounds redundant with some parts of the Introduction. I would recommend to rewrite some parts to make it shorter.Author Response
Reviewer 1 Feedback:
We thank Reviewer 1 for the expert review of this manuscript. We are encouraged by the Reviewer’s complementary feedback with regards to the pathophysiological significance of investigating ADAMTS-5 blockade in muscular dystrophy. We hope that our initial study provides strong rationale for future mechanistic studies into how ADAMTS-5 blockade may improve the function of dystrophic muscles.
- Have the authors explored the versicanase activity of ADAMTS-1 or ADAMTS-15 in the course of their experiments?
Response: It would be difficult to specifically measure the versicanase activity of ADAMTS-1 and ADAMTS-15 (or the other ADAMTS versicanases) in mdx EDL muscles following ADAMTS-5 blockade, because all of these proteases produce the DPEAAE versikine fragment. In follow up studies, it would be worthwhile to measure the protein levels of ADAMTS isoforms as this would give an indication of a potential compensatory response.
- Fig 1. To better fit with the legend and the main text, pictures corresponding to ADAMTS-1 should be shown on the left part of the figure, and those corresponding to ADAMTS-15 on the right side.
Response: We have revised Figure 1 as suggested by Reviewer, and agree that fit between the figure and text is much improved.
- Fig 4C. Loading control should be indicated in both the figure and the legend.
Response: As described in the methods, we have normalised our gels to total protein. Accordingly, have labelled the stain free gel to indicate its function as a loading control.
- Discussion is extensive and it sounds redundant with some parts of the Introduction. I would recommend to rewrite some parts to make it shorter.
Response: As suggested by Reviewer 1, 2 and the Editor, we have revised the manuscript to tighten up the Introduction and the Discussion the discussion has been extensively reworked and shorted by 250 words. Major edits are in indicated in red font.
We have also taken care to remove redundancy in content from the Results section.

Reviewer 2 Report
The study by Addinsall et al demonstrates the effect of inhibiting ADAMTS-5 using a monoclonal antibody on muscle function in the Duchenne muscular dystrophy mouse model, mdx. It is a new interesting avenue of treating this mouse model and as such the paper presents a compelling case.
However, there are some major concerns that needs to be addressed.
The introduction and discussion should be more to the point and repetition should be avoided. A good example is adamts-5 knockdown in zebrafish and its effect on shh, sonic hedgehog which is mentioned in the introduction page 3, line 92-95, discussion twice on page 16, line 470-2 and page 17, line 511-513.
Considering the effect of ADAMTS-5, it is peculiar that a group of C57BL/6 mice with the same age as the treated mdx mice was not added as a control cohort in order to demonstrate the baseline of ADAMTS-5 and versikine and whether these in any material way was involved in mouse development since these are fairly young and analyzed while still in a steep growth phase.
In the contractile function testing, it should be specified that testing was exclusively done using twitches. Was tetanic stimulation attempted? A force-frequency relationship was established, however, if only twitches were used, it is not a force-frequency relationship, but only a difference in length of that single stimulus. This would be duty cycle in train stimulation, which also is an important factor but does not replace a true force -frequency relationship. Usually a force-frequency relationship involves trains (common duration 100-500ms) as the frequency has a large impact on the fusion of single pulses into tetani. The vast majorities of treatment studies of the mdx mouse model uses trains/tetani since these reflect the physiological use of the muscle. Twitches offer a limited view of muscle function since it is a response to a single pulse and the paper should be careful in interpretation of the contraction data.
There is an issue with data units in figure 5A or table 2 and 3 since all demonstrate force as kN/m2, which is specific force, not absolute force. It appears that the graphs 5A and 5C demonstrate absolute force rather than specific force, otherwise the unit kN/m2 makes no sense at all (and the values fits with absolute force commonly achieved for twitches). It is then obvious to question what data graphs 5B and 5D are based on, specific or absolute force?
In terms of treatment, absolute force is irrelevant, only the normalized specific force is of interest, and in table 2 and 3 there is no significant difference between treated and untreated EDL and soleus, which means that the treatment has no functional effect.
Another major outcome factor is changes in regeneration. Fundamentally “improvements” in regeneration can be interpreted in two ways, either a true improvement of the degeneration-regeneration cycle is taking place, or much worse, the regeneration program is being impeded, which is detrimental to muscle function.
In this study desmin positive fibers drop, a sign of receding regeneration, but there is a trend of increased levels of myogenic factors at the mRNA level (I would disagree with the trend as the p-values are a bit high for that). The CNFs I agree is not demonstrating any difference, not can it be expected to do, as CNF’s can stay central for many months. Instead embryonic myosin can be used since this is expressed typically 1-3 days after initiation of regeneration, then followed by neonatal myosin which stays for 1-3 weeks giving a picture of immediate and ongoing regeneration. Since muscle are postmitotic, the only nuclei that divides are satellite cells, and Ki-67 is a good marker for satellite cell activation, as is myoD and myogenin as accompanying markers.
The discussion needs to be more focused on the effect or absence of effect of ADAMTS-5 based on the results achieved. At this point there is no histopathological, biochemical or functional difference between treated and untreated animals.
The title of the paper needs to be changed to reflect the absence of functional changes, since absolute changes in muscle force does not constitute a functional change if the muscles got bigger and would not be accepted as an outcome measure in any preclinical trials.
Minor comments:
In general, the data is well presented, however, for ease of understanding figure 2 should be changed to individual channels and a merged channel, since the use of phase contrast makes little sense on muscle sections unless morphological or ultrastructural features should be highlighted.
Author Response
Reviewer 2 Feedback:
We thank Reviewer 2 for their expert and careful review. We are hearted to read that our initial study on ADAMTS-5 blockade is considered by the reviewer to be an interesting avenue of treating the pathology of dystrophic mdx mice. We hope that the publication of this manuscript will lay groundwork for future, more mechanistically driven research.
- The introduction and discussion should be more to the point and repetition should be avoided. A good example is adamts-5 knockdown in zebrafish and its effect on shh, sonic hedgehog which is mentioned in the introduction page 3, line 92-95, discussion twice on page 16, line 470-2 and page 17, line 511-513.
Response: We have carefully revised the Introduction and the Discussion to avoid repetition of content; especially with regards to Shh signalling and the role of ADAMTS versicanases in myogenesis. Major edits have been highlighted in red font.
- Considering the effect of ADAMTS-5, it is peculiar that a group of C57BL/6 mice with the same age as the treated mdx mice was not added as a control cohort in order to demonstrate the baseline of ADAMTS-5 and versikine and whether these in any material way was involved in mouse development since these are fairly young and analyzed while still in a steep growth phase.
Response: For this first initial study, we focused on mdx mice because of the high upregulation of versican and ADAMTS-5 in these mice. In 2013, we published a manuscript in JBC examining the role of ADAMTS-5 and -15 in myoblast fusion. Wherein we reported a down regulation of Adamts versicanase mRNA transcripts during postnatal growth and a low level of versican remodelling in wild type muscles at 3 weeks of age (PMID: 23233679). Therefore, we chose to focus on mdx mouse muscles for this initial research. This is acknowledged in the Discussion:
Discussion: Indeed, at E13.5, intense versikine immunoreactivity was observed in developing hindlimb muscles from wild type and ADAMTS-5‑/‑ knockout mice. Whereas, at 10 days and 3 weeks of age, versikine was detected in hindlimb muscles from wild type mice but not ADAMTS-5‑/‑ knockout mice [21]. Thus, in healthy muscles compensatory versican processing by other ADAMTS versicanases may occur during developmental myogenesis, but not postnatal growth. This fits with the dramatic downregulation of Adamts-5, -1, -4 and -15 mRNA transcripts in hindlimb muscles from wild type during postnatal growth (encompassing 10 days to 3 weeks of age to adulthood) [21].
Having found an initial effect of ADAMTS-5 blockade in dystrophic hindlimb muscles, we concur with the reviewer that it would be valuable to iinclude C57BL/10 wild type mice in follow-up mechanistic studies.
- Questions and clarifications regarding contractile function testing methodology, results presentation and data interpretation:
- a) In the contractile function testing, it should be specified that testing was exclusively done using twitches. Was tetanic stimulation attempted?
Response: We used twitch stimulation to obtain optimal length and then a force-frequency relationship was used to determine the maximal isometric force producing capacity of EDL and soleus muscles.
In text – Methods (Lines 172-189): A series of 1 Hz isometric twitch contractions were used to determine optimal muscle length (Lo). Following 4 min of rest, the maximum isometric tetanic force (Po) production was determined from the plateau of the force frequency curve (FFC). The EDL muscles were stimulated at 10, 30, 50, 60, 80, 100 and 120 Hz and soleus at 10, 20, 30, 50, 60, 80 and 100 Hz, with 2 min rest between stimulations. Muscles were again rested for 4 min. To assess tolerance to repeated contractile activity and as an indicator of fatigability, muscles were stimulated sub-maximally (60 Hz) once every 5 s for 4 min [38]. Recovery of Po was determined by stimulating the muscles at 60 Hz at 2, 5, and 10 min post fatigue testing.
Following the completion of contractile function testing, muscles were trimmed of tendons, weighed and snap frozen in liquid nitrogen for gene expression analysis. Contralateral hindlimb muscles were also excised, embedded in OCT, and frozen in thawing isopentane. Muscle cross-sectional area was determined by dividing the muscle mass by the product of optimum fibre length (Lf) and 1.06 mg·mm-3, the density of mammalian muscle. Lf was determined by multiplying Lo by previously determined Lo/Lf ratios; 0.44 for the EDL and 0.71 for the soleus [39]. Since Po is dependent upon muscle size, Po values were normalised for muscle cross-sectional area and expressed as specific force (sPo; mN·mm-2) for the force frequency curve. To assess fatiguability and force recovery, data from the fatigue protocol were normalised to the first 60 Hz contraction and expressed as a percentage.
In text – Results (Lines 342-50): The ex vivo 1 Hz twitch force (Pt), time to peak tension (TPT), and ½ relaxation time (½ RT) did not significantly differ between EDL or soleus muscles from mdx mice treated with ADAMTS-5 mAb or the IgG control. When twitch force (Pt) was normalised to muscle cross-sectional area, ADAMTS-5 blockade increased the force output (sPt) in response to 1 Hz stimulation in EDL (P < 0.05), but not soleus muscles (Tables 2 and 3, respectively). This indicates that ADAMTS-5 blockade may specifically increase muscle strength in fast dystrophic hindlimb muscles. Whilst having no significant effect on muscle relaxation or Ca2+ regulation in either fast or slow dystrophic hindlimb muscles, as TPT and ½ RT are determined by myosin heavy chain composition, Ca2+ sensitivity, and SERCA isoform and function [47].
- b) … Usually a force-frequency relationship involves trains (common duration 100-500ms) as the frequency has a large impact on the fusion of single pulses into tetani….
Response: We have edited the methods to clarify the train duration used for EDL and soleus muscles. Our contractile function methods are in line with TREAT-NMD guidelines (http://www.treat-nmd.eu/downloads/file/sops/dmd/MDX/DMD_M.2.2.005.pdf).
In text – Methods (Lines 167-169): EDL and soleus muscles were stimulated by supramaximal square wave pulses of 350 ms and 1200 ms in duration, respectively, delivered by two platinum electrodes that flanked the length of the muscle.
- There is an issue with data units in figure 5A or table 2 and 3 since all demonstrate force as kN/m2, which is specific force, not absolute force. It appears that the graphs 5A and 5C demonstrate absolute force rather than specific force, otherwise the unit kN/m2 makes no sense at all (and the values fits with absolute force commonly achieved for twitches). It is then obvious to question what data graphs 5B and 5D are based on, specific or absolute force?
Response: Apologies for including the incorrect units in our graphs, an error we missed in editing. All data related to specific force (sPo and sPt) are presented as mN/mm2, as is standard for published work from our group.
The force-frequency data presented in Figure 5A and C is expressed as specific force (sP0). We have also revised the twitch data in Table 2 (EDL) and Table 3 (soleus) to include not just twitch force Pt but also twitch force normalised to muscle size (sPt).
It is worth noting that ADAMTS-5 blockade increased the strength of EDL muscles as indicated by increased sPt and upward shift of the sPo force frequency curve.
The findings presented in Figure 5B and D are the fatigue and recovery data. These data are expressed as a percentage of the sPo output for the first contraction (at 60 Hz) from the 4 min fatigue protocol. During the fatigue protocol, muscles were stimulated submaximally (at 60 Hz) every 5 s and force recovery was assessed at 2, 5 and 10 min post.
We have revised the methods and corresponding results sections for increased clarity with regards to the twitch data (including sPt). the force frequency data (sPo ), and the fatigue/recovery data. Hopefully, these edits improve the presentation of our findings.
In text – Methods (Lines 187-9): To assess fatiguability and force recovery, data from the fatigue protocol were normalised to the first 60 Hz contraction and expressed as a percentage.
In text – Results (Lines 351-60): In concordance with the 1 Hz twitch data, in EDL muscles ADAMTS-5 blockade increased the isometric strength normalised to muscle cross-sectional area (sPo), as indicated by an upward shift in the 1 to 120 Hz force frequency curve (P < 0.001; Fig. 5A). Whereas, in slow soleus muscles, ADAMTS-5 blockade had no significant effect on isometric muscle strength (sPo; Fig. 5C). Following assessment of isometric strength, muscle endurance was assessed. During the 4 min of intermittent, submaximal stimulation at 60 Hz, EDL muscles from mdx mice treated with the ADAMTS-5 mAb fatigued less than IgG control treated littermates (P < 0.05), and force recovery was enhanced (P < 0.05; Fig. 5B). In soleus muscles, ADAMTS-5 blockade had no significant effect on the rate of fatigue or force recovery (Fig. 5D). Altogether, these data demonstrate that the positive effects of ADAMTS-5 blockade on the contractile function of dystrophic hindlimb muscles is fiber type specific.
- Questions regarding ADAMTS-5 blockade and whether it affect regeneration in dystrophic EDL muscles.
- Another major outcome factor is changes in regeneration. Fundamentally “improvements” in regeneration can be interpreted in two ways, either a true improvement of the degeneration-regeneration cycle is taking place, or much worse, the regeneration program is being impeded, which is detrimental to muscle function. In this study desmin positive fibers drop, a sign of receding regeneration, but there is a trend of increased levels of myogenic factors at the mRNA level (I would disagree with the trend as the p-values are a bit high for that).
Response: Given that ADAMTS-5 blockade improved the specific force output of EDL muscles (indicating an increase in strength), we would argue the changes seen in desmin positive muscle fibres would suggest a positive effect on muscle regeneration.
Nonetheless, in response to Reviewer 2 and Editor feedback, in revising the manuscript, we have been more careful and circumspect with the interpretation of our data, especially the gene expression data.
In text – Results (Lines 404-9): A non-significant increase in the mRNA transcript abundance of the myogenic regulatory factors Myf5 and Myogenin was also observed in EDL muscles from mdx mice treated with the ADAMTS-5 mAb (Fig. 7D-E; P = 0.140 and P = 0.116, respectively). The desmin immunohistochemistry, and perhaps the myogenic regulatory factor gene expression data, offer preliminary in vivo evidence that ADAMTS-5 blockade may have the potential to modulate regenerative myogenesis in dystrophic muscles.
- The CNFs I agree is not demonstrating any difference, not can it be expected to do, as CNF’s can stay central for many months. Instead embryonic myosin can be used since this is expressed typically 1-3 days after initiation of regeneration, then followed by neonatal myosin which stays for 1-3 weeks giving a picture of immediate and ongoing regeneration. Since muscle are postmitotic, the only nuclei that divides are satellite cells, and Ki-67 is a good marker for satellite cell activation, as is myoD and myogenin as accompanying markers.
Response: We agree with reviewer that more detailed analyses of how ADAMTS-5 blockade might affect muscle regeneration would be very valuable, and we have acknowledged this in the discussion.
To complete additional immunhistochemical analyses is not feasible in the time provided for manuscript revision. Furthermore, a limitation of our study is the use of male and female mdx mice, which was a necessity with regards to animal ethics and acceptable for initial study. However, the use of male and female mdx mice increases biological noise and detailed mechanistic studies should to be conducted on a new cohort of male mdx mice and include C57BL/10 wild type controls.
In text – Discussion (Lines 495-504): To begin to assess the effects of ADAMTS-5 blockade and muscle repair, desmin positive muscle progenitor cells and desmin positive, newly regenerated muscle fibers were quantified in EDL cross-sections. The reduction in the number of desmin positive muscle fibers, but not desmin positive progenitor cells, suggests that ADAMTS-5 blockade may improve the efficacy of regenerative myogenesis, which in turn would account for the increase in EDL muscle strength. This interpretation needs to be carefully interrogated in follow-up studies using more sensitive markers of regenerative efficacy. Specifically, immunohistochemical staining for muscle fibres expressing embryonic and neonatal myosin to assess initiation of regeneration at 1 to 3 days post-damage and ongoing regeneration 1 to 3 weeks post injury, respectively. This should be supported by an assessment of satellite cell proliferation and myoblast differentiation.
- The discussion needs to be more focused on the effect or absence of effect of ADAMTS-5 based on the results achieved. At this point there is no histopathological, biochemical or functional difference between treated and untreated animals … The title of the paper needs to be changed to reflect the absence of functional changes, since absolute changes in muscle force does not constitute a functional change if the muscles got bigger and would not be accepted as an outcome measure in any preclinical trials.
Response: The discussion has been revised to carefully discuss the effects of ADAMTS-5 blockade on EDL and soleus muscle function. Based on our findings, ADAMTS-5 blockade improved the strength and endurance of EDL muscles, and this may be due to its potential effects on regenerative myogenesis.
We believe that the title accurately reflects the content of the manuscript, and this was not a concern flagged by the Editor or Reviewer 1.
- Minor Comment - In general, the data is well presented, however, for ease of understanding figure 2 should be changed to individual channels and a merged channel, since the use of phase contrast makes little sense on muscle sections unless morphological or ultrastructural features should be highlighted.
Response: The ADAMTS-5, desmin and CD68 antibodies are all raised in rabbits, to assess co-localisation of ADAMTS-5 with inflammation or regenerative myogenesis serial muscle cross-sections were used. The merged phase image provides morphological evidence of co-localisation.
We have edited the methods and legend for Figure 2 to more clearly explain this.
In text – Methods (Lines 194-8): Serial sections were used for the co-localization experiments because the anti-ADAMTS-5, anti-CD68 and anti-desmin antibodies were all raised in the same species (rabbit). To confirm co-localization based on tissue morphology phase images were captured and overlaid with the corresponding fluorescent images.
In text – Figure Legend 2 (Lines 290-9): FIGURE 2 ADAMTS-5 is highly expressed in regions of regeneration and inflammation in dystrophic muscles. ADAMTS-5 immunoreactivity (red), desmin or CD68 immunoreactivity (green) and nuclei (blue) in serial cross-sections. Phase contrast images overlaid the fluorescent signal provide morphological evidence of ADAMTS-5 and desmin or CD68 co-localisation.

Round 2
Reviewer 2 Report
The authors have addressed my concerns adequately and overall the manuscript has improved considerably.
Author Response
"The authors have addressed my concerns adequately and overall the manuscript has improved considerably."
This is the best sort of feedback one can hope for. Thank you very much.